# Docking studies and molecular dynamics simulations of the binding characteristics of waldiomycin and its methyl ester analog to *Staphylococcus aureus* histidine kinase

**Awwad Radwan**[1,2]*, **Gamal M. Mahrous**[1]

**1** Kayyali Chair, College of Pharmacy, King Saud University, Riyadh, Saudi Arabia, **2** Department of Pharmaceutical Organic Chemistry, Faculty of Pharmacy, Assiut University, Assiut, Egypt

* dhna_2001@hotmail.com

**Data Availability Statement:** All relevant data are within the paper and its Supporting Information files.

## Abstract

Bacterial histidine kinases (HKs) are considered attractive drug targets because of their ability to govern adaptive responses coupled with their ubiquity. There are several classes of HK inhibitors; however, they suffer from drug resistance, poor bioavailability, and a lack of selectivity. The 3D structure of *Staphylococcus aureus* HK was not isolated in high-resolution coordinates, precluding further disclosure of structure-dependent binding to the specific antibiotics. To elucidate structure-dependent binding, the 3D structure of the catalytic domain WalK of *S. aureus* HK was constructed using homology modeling to investigate the WalK–ligand binding mechanisms through molecular docking studies and molecular dynamics simulations. The binding free energies of the waldiomycin and its methyl ester analog were calculated using molecular mechanics/generalized born surface area scoring. The key residues for protein–ligand binding were postulated. The structural divergence responsible for the 7.4-fold higher potency of waldiomycin than that of its ester analog was clearly observed. The optimized 3D macromolecule–ligand binding modes shed light on the *S. aureus* HK/WalK–ligand interactions that afford a means to assess binding affinity to design new HK/WalK inhibitors.

## Introduction

Protein kinases (PKs) catalyze the phosphorylation of other proteins and thus play crucial roles not only in cell regulation but also in signal transduction. Cell dysregulation is often associated with a disease. Thus, PKs are considered as important targets in drug discovery programs that seek to alter their function [1,2]. Particularly, histidine kinases (HKs) are responsible for sensing conditions that allow microbes to adopt a pathogenic lifestyle through the expression of virulence factors. Thus, the inhibition of factors that are distinctly unique to the bacterial cell and express virulence of pathogenic bacteria offers a chance for specific interference from the host biochemical processes. Nowadays, it is clear that virulence is an adaptive genetic response involving the genes encoding the induction of virulence factors [3,4]. This

**Funding:** A A Radwan. RG-1435-080. Deanship of the Scientific Research at King Saud University. dsrs.ksu.edu.sa/en The funders had no role in study design, data collection and analysis, decision to publish, or preparation of the manuscript.

**Competing interests:** The authors have declared that no competing interests exist.

response implies that pathogenic microbes can sense when it is in an invasion state. Sensing of environmental stimuli occurs through two-component signaling (TCS) systems that appears almost ubiquitously in bacteria. Bacteria respond to extracellular changes through TCS pathways, which are mediated by HKs and are absent in humans [5]. The adaptive and response ability of bacteria is vital for survival in severe environments that they encounter [6].

The *Staphylococcus aureus* bacteria TCS system comprises HK protein, containing a membrane-bound sensor histidine kinase (WalK), and a response regulator (RR) protein, containing a conserved DNA-binding domain (WalR) [7]. WalK is a membrane-linked HK that consists of two domains. One domain is a catalytic or ATP-binding domain and the second is a dimerization domain that includes a conserved phosphorylated histidine residue.

HKs are activated as a result of the interaction of external stimuli with its extracellular signaling domain. The activated HKs bind ATP at its catalytic domain WalK, which catalyzes the transfer of the a-phosphate group of ATP to a conserved histidine residue on the HK dimerization and histidine phosphotransfer (DHp) domain. The phosphoryl group transfers from the conserved phosphorylated histidine on a HK to aspartic acid conserved in an RR domain protein, the HK's associated signaling partner. The RR proteins harbor both the RR domain and the DNA-binding domain. The phosphorylation of RR results in its activation and subsequently triggers it to bind to DNA and stimulate gene transcription, producing a response to the extracellular stimulus (Fig 1) [5,8]. Antibacterial activity can be afforded by inhibiting the microbial HK using ATP competitive inhibitors [9,10].

The TCS-WalK/WalR system that is unique to gram-positive bacteria, including *S. aureus* [11], *Bacillus subtilis* [12], and *Enterococcus faecalis* [13], is crucial for survival and is attracting growing interest as a novel target of antibacterial agents [14]. Representative inhibitors, RWJ-49815, closantel, and tetrachlorosalicylanilide, have been reported as active inhibitors against several bacterial TCS transduction systems [15–17]. Unfortunately, few TCS inhibitors have been reported against *S. aureus*, a major pathogen with a high virulence factor. The amino acid sequence of the WalK/WalR TCS transduction histidine protein kinase of *S. aureus* (strain Mu50/ATCC 700699) ARLS_STAAM (Q7A2R7) [18] is well defined; however, the 3D structure is unavailable in the protein databank (PDB) (**https://www.rcsb.org/**).

For case in which the atomic structures of enzyme macromolecules are absent, homology modeling is a powerful tool to understand the general fold of the enzyme, catalytic site, and conserved residues, in addition to the substrate and inhibitor binding sites. This information is crucial for structure-based drug design approaches for the development of potential enzyme inhibitors for therapeutic application [19]. In addition, waldiomycin and its methyl ester analog, produced by *Streptomyces* sp., were reported as inhibitors of *S. aureus* HK with $IC_{50}$ values of 10.2 and 75.8 μM, respectively, and are effective as antibacterial agents against *S. aureus* species [20]. This information is also helpful for the ligand-based drug discovery of novel lead compounds with potential activity against *S. aureus* bacteria.

In this study, we performed *in silico* modeling to explore the mode of inhibition of the *S. aureus* HK/WalK enzyme by waldiomycin and its methyl ester analog. The study includes homology modeling, molecular docking, molecular dynamics (MD) simulations, and molecular mechanics/generalized born surface area (MM/GBSA) calculations. The calculations were performed on the *S. aureus* kinase structure (WalK catalytic domain) to investigate the macromolecule–ligand complex and predict the 3D structure of *S. aureus* HK/WalK. The *S. aureus* HK/WalK conformation changes persuaded in the MD simulations and the ligand binding free energies of waldiomycin and its ester analog, as calculated using the MM/GBSA method, confirm the dependencies of site activity on the ligands' electrostatic and hydrophobic properties. Lys100 was found to be a crucial residue for the 7.4-fold higher potency of waldiomycin than that of its ester analog against *S. aureus* HK/WalK.

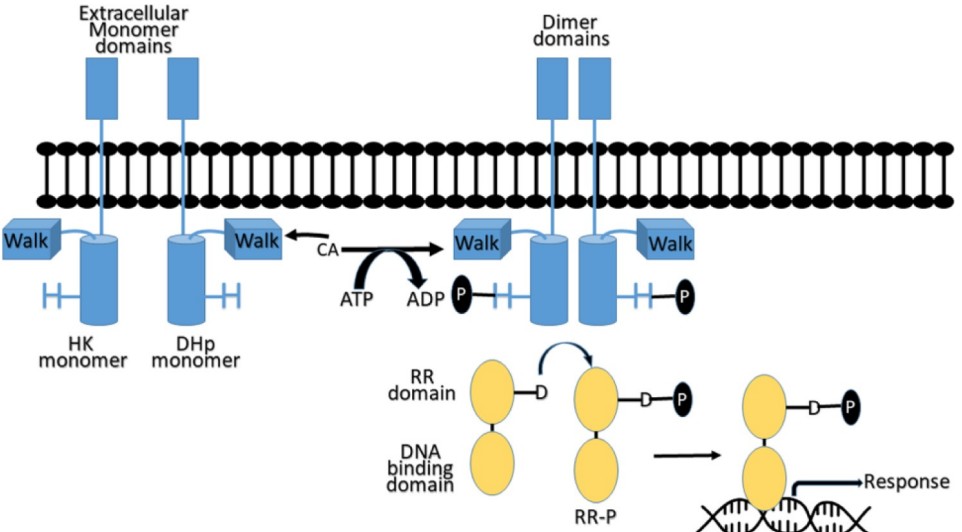

**Fig 1. Two-Component Signaling (TCS) system pathway.** HK (blue), histidine kinase; RR (beige), response regulator; H, conserved histidine residue; D, conserved aspartic acid residue.

## Methods

### Overall workflow

The schematic representations of the overall study workflow are illustrated in Fig 2. Firstly, the *S. aureus* kinase protein sequence was used for searching in the PDB. The BLAST [21] program was utilized to select a template protein possessing the highest similarity. An initial *S.*

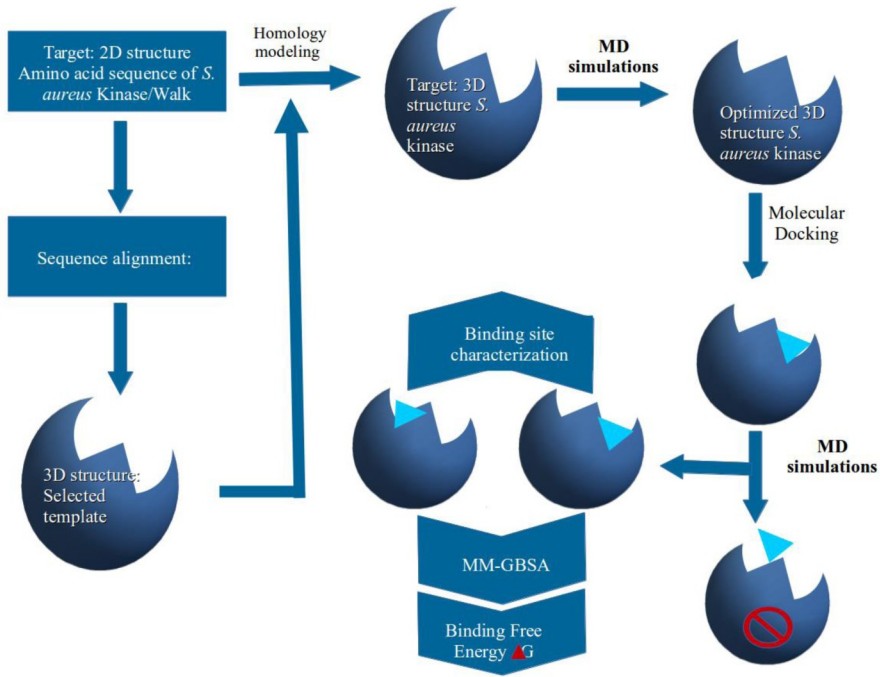

**Fig 2. Sketch representation of the overall workflow.**

*aureus* kinase 3D structure was built by means of homology modeling followed by structure optimization using an MD simulation. The optimized macromolecule structure was then used for docking analyses. Because of their almost similar structures and the large difference in their potencies as *S. aureus* HK/WalK inhibitors, waldiomycin and its ester analog [20] were selected and their 3D structures were also built, optimized, and then docked into the homology-built and optimized *S. aureus* HK/WalK structure. The *S. aureus* HK/WalK–ligand complexes were further subjected to optimization *via* MD simulations. Lately, the MD simulation trajectories were used as inputs for the MM/GBSA calculation of binding free energies of the ligands and investigation of their binding mechanism.

## Homology modeling

The homology modeling of *S. aureus* HK/WalK secondary structure was performed to develop the initial structure needed for the MD simulation study. Briefly, the BLAST sequence database search module was utilized to obtain the X-ray structures similar to the target sequence Q7A2R7 [22] and sort them according to similarity percent. Using the MODELLER program, the top-scoring PDB entry was suggested as a template for alignment of the target Q7A2R7 to build the homolog model. During the modeling process, the position of the crystallized ligand was kept in its original position for rebuilding the binding cavity. The obtained homology model was checked for missing atoms and/or residues using the PROCHECK program [23]. Gaps and missing coordinates were repaired using the MODELLER program [24] and the resulting structure was used as the initial structure in the next MD simulation optimization process.

## Protein preparation

The AMBER18 program [25] was utilized to perform the structural minimization and MD simulations using the ff14SB force field amino acid parameters. The tleap module of Amber-Tools18 was used to add the complementary hydrogen atoms that could not be detected by X-ray crystallography. The protein molecule was explicitly solvated with water molecules and sodium counter-ions were also added to obtain a neutral system. A water box with 10 Å thickness was created using the TIP3P water solvent model to completely surround the protein molecule. Simulation was performed using a periodic boundary condition. Long-range electrostatic interactions were processed using the particle–mesh Ewald protocol [26] and a cut-off value of 8 Å was used for non-bonding interactions. Initial minimization of the water molecules and counter-ions was performed using 1000-cycle minimizations; subsequently, the whole system was minimized using 1000-cycle minimizations. Then, a system temperature increase to 298.15 K was performed and MD simulations at 298.15 K were executed with the fixed protein coordinates. The equilibrations were performed under the constant normal temperature and pressure (NPT) ensemble for 170 ps. In the first 20 ps, the waters and counter-ions were equilibrated, while the solutes were restrained. In the next 50 ps, the side chains of the protein amino acids were relaxed, then all the restraints were removed in the last 100 ps. Lastly, MD simulations of 10 ns were processed at 1 atm and 298.15 K under the NPT ensemble with a time step of two femtoseconds (fs). The covalent bonds with hydrogen atoms were constrained by the use of the SHAKE algorithm [27] and Langevin dynamics were used to control the system temperature. Throughout the MD simulations, all of the atom coordinates of the system were saved every 1 ps.

## Ligand preparation

Waldiomycin and its methyl ester **1, 2** were reported as inhibitors of *S. aureus* HK with $IC_{50}$ values of 10.2 and 75.8 $\mu$M, respectively [20]. The chemical structures and compound names

are given in Fig 3. The antechamber module [28,29] in the AmberTools18 program was used to generate the force field parameters of the ligands. The general AMBER force field was applied to the ligands, and the Leap program was used to generate the topology and parameter files.

## Docking study

The docking procedure was performed using the Dock6.4 program [30] on an ubunto 14.2-supported Dell 5000 workstation (Processor: Intel(R) Core(TM) i7-5500U CPU @ 2.40GHz with a 64-bit operating system, ×64-based processor, memory of 7.7 GiB). The docking method was validated using 5C93 proteins and co-crystalized ligand ACP using the default parameters, except with a sphere of 10 Å diameter around the binding site center. The co-crystallized ligand ACP was docked in the original binding pocket of the protein structure. The docking protocol was validated according to the root-mean-square deviation (RMSD) of the docking pose relative to the ACP X-ray structure.

The refined tertiary structures of *S. aureus* HK/WalK and HK/WalK inhibitor compounds **1, 2** were prepared in their physiological protonation states. Compounds **1** and **2** were docked using the same settings for the validation process of the docking program. The resulting solutions were clustered based on the heavy atom RMSD values (1 Å). Finally, the top-scoring binding pose was taken into consideration for ligand–receptor interactions and was selected for further refinement using MD simulations.

## MD simulations of complexes

MD simulations were performed for the HK–ligand complexes obtained via docking using the same procedure explained in the MD optimization described above, including system minimization, heating, and equilibration. MD simulations were performed for a time period of 35 ns for each macromolecule–ligand complex. The atom coordinates in the complex system were saved at 1-ps time intervals throughout the simulations. Using the initial structures of the MD simulation as the reference structures, the CPPTRAJ module of AmberTools18 was used for determining RMSDs to validate the convergence of the MD simulation processes. Structural flexibilities of the protein were estimated by calculating root-mean-square fluctuations (RMSFs). The trajectories during last 20 ns of equilibration MD simulations were used for calculating the RMSF values, whereas the average structures of the final 20 ns trajectories were used as the reference structures.

**Hydrogen bond analysis.** The CPPTRAJ module in Amber18 was used to analyze hydrogen bonding between the ligand and the surrounding amino acid residues. Hydrogen bonds with high occupancy were observed during the last 8.0 ns of the simulations for the two ligand complex systems.

**Pairwise per-residue free energy decomposition.** Molecular mechanics Poisson–Boltzmann surface area (MM/PBSA) pairwise decomposition was used to calculate the interaction energy of the residues involved in high-occupancy hydrogen bonding to the ligand during the last 8 ns of the simulation.

## Binding free energy calculation

The MM/GBSA method [31] was used to calculate the binding free energy (ΔGbinding), Eq (1) for receptor–binder complex systems. One thousand snapshots were taken at 20–30 ns time intervals throughout the MD simulation trajectory to compute the MM/GBSA free energy difference. For each snapshot, Eq (1) was used to calculate the binding free energy of

**Fig 3. Chemical structures of *S. aureus* HK/WalK inhibitors.** (1) Waldiomycin (IC$_{50}$ 10.2 μM) and (2) waldiomycin methyl ester (IC$_{50}$ 75.8 μM).

the complex system.

$$\Delta G_{binding} = G_{complex} - (G_{protein} + G_{ligand}), \tag{1}$$

$$\Delta G = \Delta G_{gas} + \Delta G_{solv} - T\Delta S, \tag{2}$$

$$\Delta G_{gas} = \Delta E_{electrostatic} + \Delta E_{vdw}, \tag{3}$$

$$\Delta G_{solv} = \Delta G_{GB} + \Delta G_{SA}, \tag{4}$$

$$\Delta G_{SA} = \gamma \times SASA + b, \tag{5}$$

where $G_{complex}$, $G_{protein}$, and $G_{ligand}$ are the free energies of the complex, protein, and ligand, respectively; $\Delta G_{gas}$ is the gas-phase molecular mechanics free energy, including electrostatic ($\Delta E_{electrostatic}$) and van der Waals ($\Delta E_{vdw}$) contributions; and $\Delta G_{solv}$ is the solvation free energy, which includes polar ($\Delta G_{GB}$) and nonpolar ($\Delta G_{SA}$) contributions.

$\Delta G_{GB}$ is the polar contribution that was estimated using the modified GB model reported by Onufriev et al. [32] using εw = 80 and εp = 1.0, while the solvent-accessible surface area (SASA) is calculated using linear combinations of pairwise overlaps [33]. The surface tension

proportionality constant (γ) was set to 0.0072 kcal mol$^{-1}$Å$^{-2}$ while the nonpolar solvation free energy for a point solute (b) was set to 0.00 kcal mol$^{-1}$. The SASA was calculated using a probe sphere radius set to 1.4 Å. The calculated free energies were used for comparisons between the waldiomycin and waldiomycin methyl esters as *S. aureus* kinase enzyme binders. The final binding free energy for the two binders was calculated as the average value from 1000 snapshots in the last 10 ns of the MD simulations.

## Results and discussion

### Structure optimization

With an identity of 36%, the sequence of *S. aureus* UID-Q7A2R7 [18] has the highest-ranked homology in the BLAST search for *E. coli* histidine kinase, 5C93 [34], on proteins in the PDB. Generally, it is assumed that two proteins topologies are usually similar once their sequences share about 30% identity [35]. Moreover, Dimitrii et al. [36] developed structures of average quality (RMSD < 4.0 Å) based on target-template sequence identity ≥ 20%. The crystal structure of 5C93 was recently obtained at a resolution of 2.52 Å [34]. Therefore, the 3D structure of *S. aureus* HK, built through a homology model to the 5C93 template structure, should be appropriate for modeling *S. aureus* HK protein binding data. The initial alignment of Q7A2R7 with the template sequence 5C93 was obtained using ClustalW. The final alignment (36% identity) used in the homology modeling and in the secondary structure prediction is shown in Fig 4. The initial homology models of *S. aureus* HK were built using MODELLER and used for further optimization.

MD simulation is usually practiced to improve homology models [37]. Using MD simulation, the initial 3D structure of *S. aureus* HK/WalK, obtained from homology modeling, was optimized in a solvent to simulate the real physiological environments. During the MD simulation, the stability of the structure was calculated by its deviance from the starting conformation in RMSD terms. Fig 5A shows the RMSD values of *S. aureus* HK/WalK protein structure through the whole MD simulation trajectory. After 7 ns, the 3D structures of HK/WalK reached a stable state where the RMSDs of the protein backbone atoms and of all atoms converged to 1.25 Å and 2.25 Å, respectively. Fig 5B shows the calculated RMSFs of the *S. aureus* HK/WalK structure generated during the MD simulation, which characterizes the mobility of individual residues. The Ramachandran plot of the optimized *S. aureus* HK/WalK structure showed that 100% of residues were located in the allowable regions (S1 Fig), higher than that of the initial homology modeled 3D *S. aureus* HK (S2 Fig), which showed 93.4% of residues in the allowed region, indicating the higher stability of the optimized *S. aureus* HK structure.

### Binding modes generated by molecular docking

Redocking of the ACP X-ray structure in its original binding site resulted in a solution pose with 0.704 Å and RMSD value S3 Fig.

To shed light on the possible binding modes of compounds **1** and **2**, a large box was defined to include all potential binding sites. Hence, the two ligands adopted the most favorable binding poses that were searched and ranked according to their docking scores. The conformation with the lowermost score, suggesting the most probable binding mode of a ligand, was suggested for additional analysis.

The two binders were efficiently docked inside the binding site of the HK/WalK structure. The binding poses of the two binders showed similar V-shaped orientations inside a binding site delineated with the side chains of residues Lys63, Lys100, Leu102, Hie103, Pro104, Leu131, Ile132, Asp135, Ile138, Lys139, Arg183, Val184, Asp185, and Asn195. The hydrophobic part of

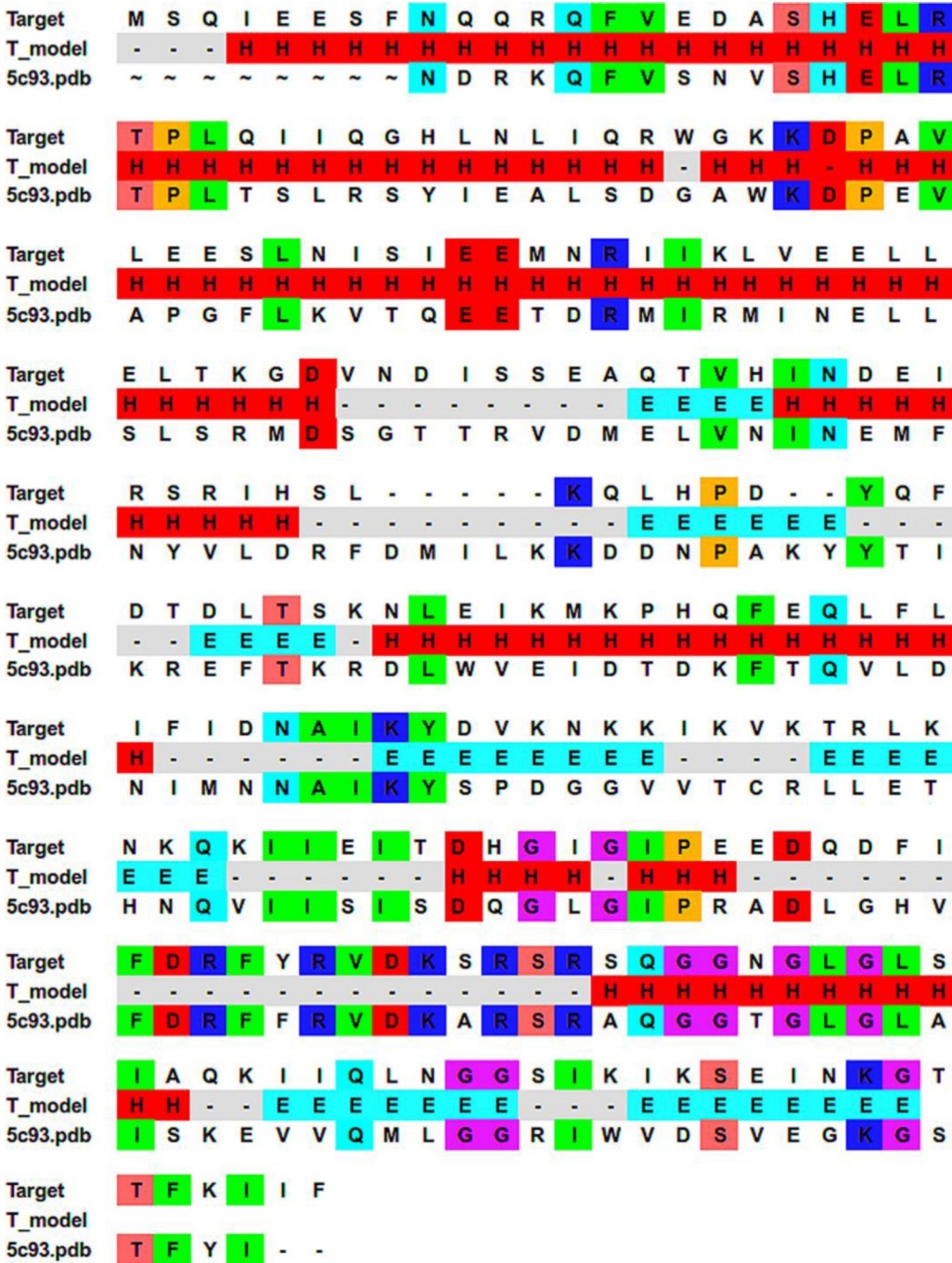

**Fig 4. _Staphylococcus aureus_ Q7A2R7 sequence (target) alignment result with template sequence (5c93).** The obtained secondary structure is shown, where H represents the helix; E represents the strand; and "-" represents the loop. Identical residues are highlighted.

the binders fits inside one binding pocket near Val184, while the carboxyl and ester groups of the two compounds fit inside a second binding pocket near Lys100 (Fig 6A and 6B).

The 10 top-scoring conformations of the waldiomycin and waldiomycin ester are shown in S4 Fig and S5 Fig mol2 files, respectively. The docking score and hydrogen bond information

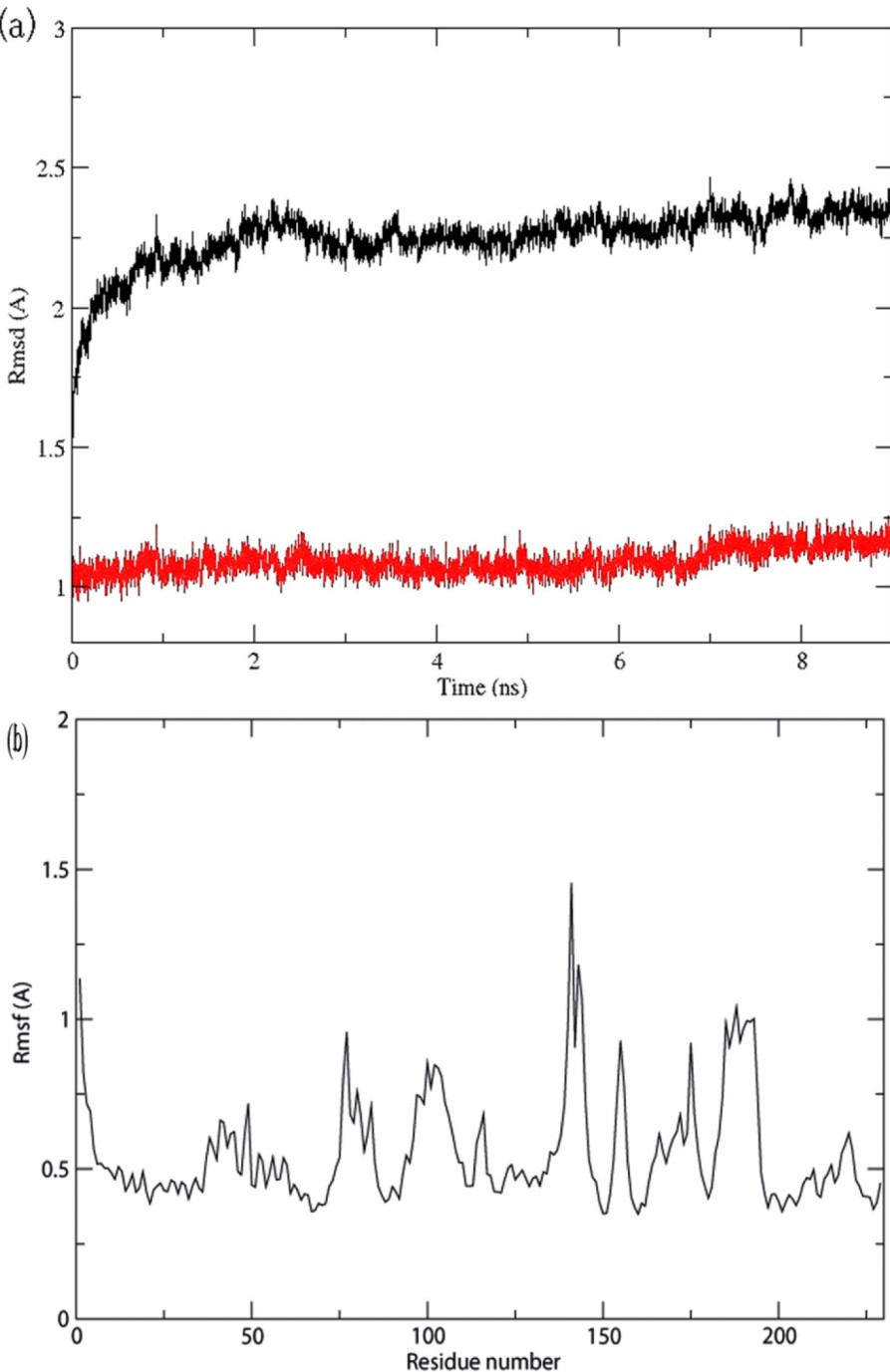

**Fig 5. Molecular Dynamics (MD) profiles of the *S. aureus* HK 3D structure optimization.** a) Root-mean-square deviation (RMSD) values (y-axis) along the time frame (x-axis) for atoms of the protein backbone (red lines) and all protein atoms (black line), and b) root-mean-square fluctuation (RMSF) values (y-axis) in the MD simulation for individual residues (x-axis).

of these 10 conformations are also listed in S1 and S2 Files, respectively. The Grid score of these binding poses is in the range of –63.629486 to –62.780089 kcal mol$^{-1}$ for waldiomycin and –66.089516 to –64.42131 kcal mol$^{-1}$ for waldiomycin methyl ester, which is incomparable

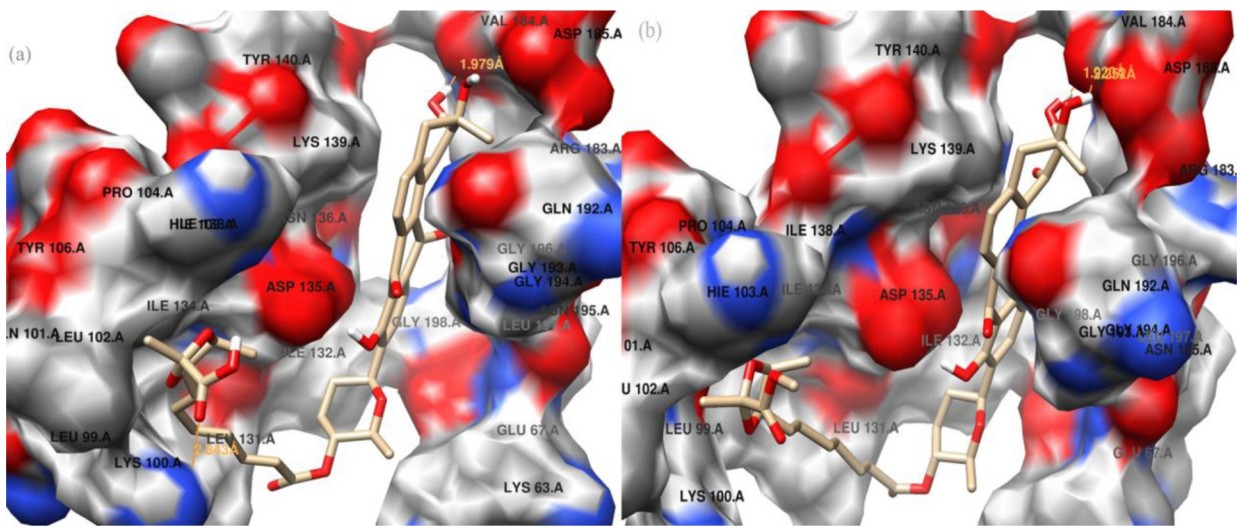

**Fig 6. Docking results of homolog modeling.** Binding conformations of waldiomycin (a) and waldiomycin methyl ester (b) at the binding site of the homolog model of *S. aureus* HK/WalK.

with the difference in their experimental inhibition of *S. aureus* HK/WalK, with IC$_{50}$ values of 10.2 and 75.8 **μ**M, respectively. The top-scoring binding poses of the two binders were selected for further investigations using MD simulation.

Although docking studies have been successfully used in calculating the binding poses of ligands for many proteins, they failed in assessing ligand binding affinity [38]. The treatment of proteins as "rigid" molecules does not consider their conformational changes during the docking of some ligands [39]. Conformational changes during protein–ligand interactions could be studied during MD simulations. The conformation changes in protein–ligand interactions have been broadly studied through MD simulation methods [40]. Considering both protein structure flexibility and protein–ligand electrostatic interactions, MD simulations

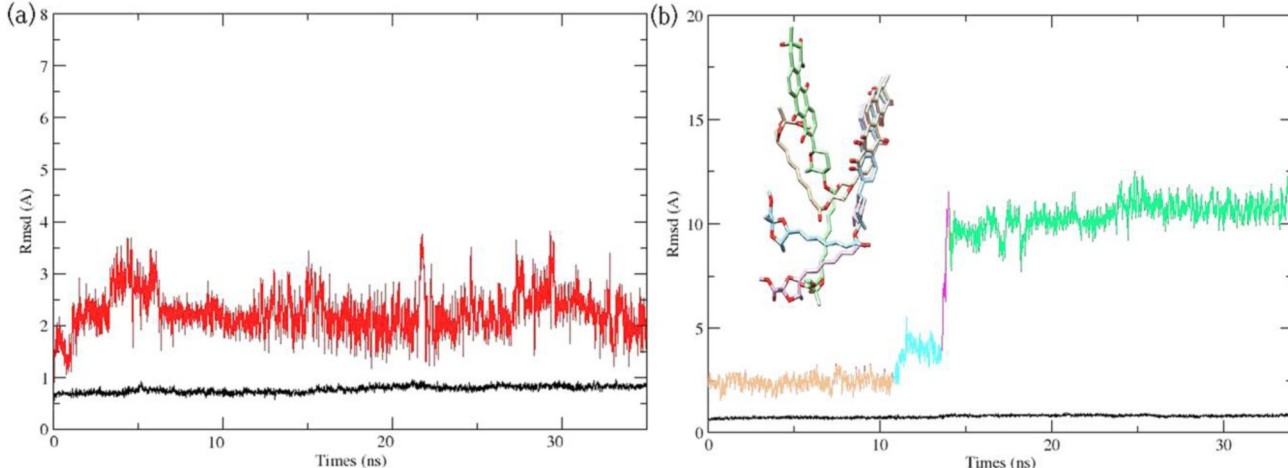

**Fig 7. Root-Mean-Square Deviation (RMSD) in Molecular Dynamics (MD) simulations.** (a) RMSD of protein backbones (black lines) and waldiomycin (red lines). (b) RMSD of protein backbones (in black lines) and of waldiomycin methyl ester (in similar color) with the four orientations of waldiomycin methyl ester through the MD simulation time intervals. 0–11 ns (beige color), 11–14 ns (cyan color), 14–14.5 ns (pink color), and 14.5–35 ns (green color).

were deemed necessary to justify the 7.4-fold potency difference in anti-*S. aureus* HK/WalK activity between waldiomycin and its ester analog and to refine the understanding of waldiomycin (ester)-HK/WalK binding modes obtained from molecular docking. Ligand (waldiomycin or its methyl ester) conformation changes were observed through all the MD simulation time frames. MD simulation was initiated from the docked minimized complex and continued for over 35 ns. Using RMSD changes during the MD simulations, the dynamic stabilities of the two complex systems were calculated and are plotted in Fig 7A and 7B.

For the complexes of *S. aureus* HK/WalK bound with waldiomycin, Fig 7A, both the waldiomycin and the protein structures were equilibrated with unobvious RMSD fluctuations through the 35 ns time frames. Interestingly, trajectory analysis revealed that the carboxyl group of waldiomycin is engaged in stable hydrogen bond with Lys100. The presence of a hydrogen bond between the Lys100 residue of HK/WalK and waldiomycin (active anti-HK) and its absence with the waldiomycin methyl ester (inactive anti-HK) suggests that Lys100 is a crucial residue for *S. aureus* HK activity.

The complexes of *S. aureus* HK/WalK bound with waldiomycin methyl ester (Fig 7B) showed unobvious RMSD fluctuations over the first 11 ns, then a gradual increase from 11–14 ns, followed by a rapid increase from 14 to 14.5 ns, and convergence from 14.5 to 35 ns. The large RMSD fluctuations suggest a large conformational change and repositioning of the waldiomycin ester inside the binding site resulting in four conformational changes (Fig 7B; beige color): holds for 0–11 ns, (7b; cyan color), holds during the 11–14 ns, (Fig 7B; pink color), holds during the 14–14.5 ns (Fig 7B; green color), and continues over the last 14.5–35 ns. Remarkably, trajectory analysis revealed that the ester group of the waldiomycin methyl ester exited the binding pocket after 14.5 ns; this conformational instability during MD simulation is unsurprising since a weak binding affinity was reported for this binder.

The superimposition of binding conformation from the final snapshots of the MD simulations is shown in Fig 8 (waldiomycin: gray; waldiomycin methyl ester: cyan) and reveals different ligand binding conformations, though the MD simulations started from docking poses with similar orientations.

The CPPTRAJ hydrogen bond analyses are listed in Table 1 and show the hydrogen bonds with high occupancy for the two simulation systems. The table shows two different hydrogen bond sets for the two complex systems. For the waldiomycin-HK/WalK complex, the amino acid residues Lys63, Ser98, Lys100, Gln128, Asp135, Lys139, and Val184 showed a hydrogen bond with the ligand. For the waldiomycin ester-HK/WalK complex, the amino acid residues, Lys63, Leu102, and His103 showed a hydrogen bond with the ligand. The hydrogen bond pattern difference between the two simulation systems is an indication of the two different binding conformations of waldiomycin and its ester analog (Fig 8).

The MM/PBSA-pairwise decomposition [41] of residue interaction energies developed an insight into the interactions between the binders and their binding sites for HK/WalK. The results demonstrated that the amino acid residues Lys100, Asp135, Lys139, and Val140 play a crucial role in effective binding interactions with waldiomycin and showed absolute decomposed energy in the range of –2.722 to –4.371 kcal mol$^{-1}$ (Table 1).

Although the MM/GBSA approach does not consider the entropy associated with the binding, it is an efficient and powerful tool for the correct ranking of ligands. Although the inclusion of the conformational entropy is crucial for calculating the absolute free energies of binding, it is not necessary for ranking the binding affinities of similar inhibitors [42].

The binding free energies of the two ligands were calculated using MM/GBSA methods according to the MD simulation trajectories from 25 to 35 ns. The total binding free energy $\Delta G_{MM/GBSA}$ and the related energy terms determined using the MM/GBSA method are provided in Table 2. As it was used in direct comparisons with $\Delta G_{MM-GBSA}$, the experimental

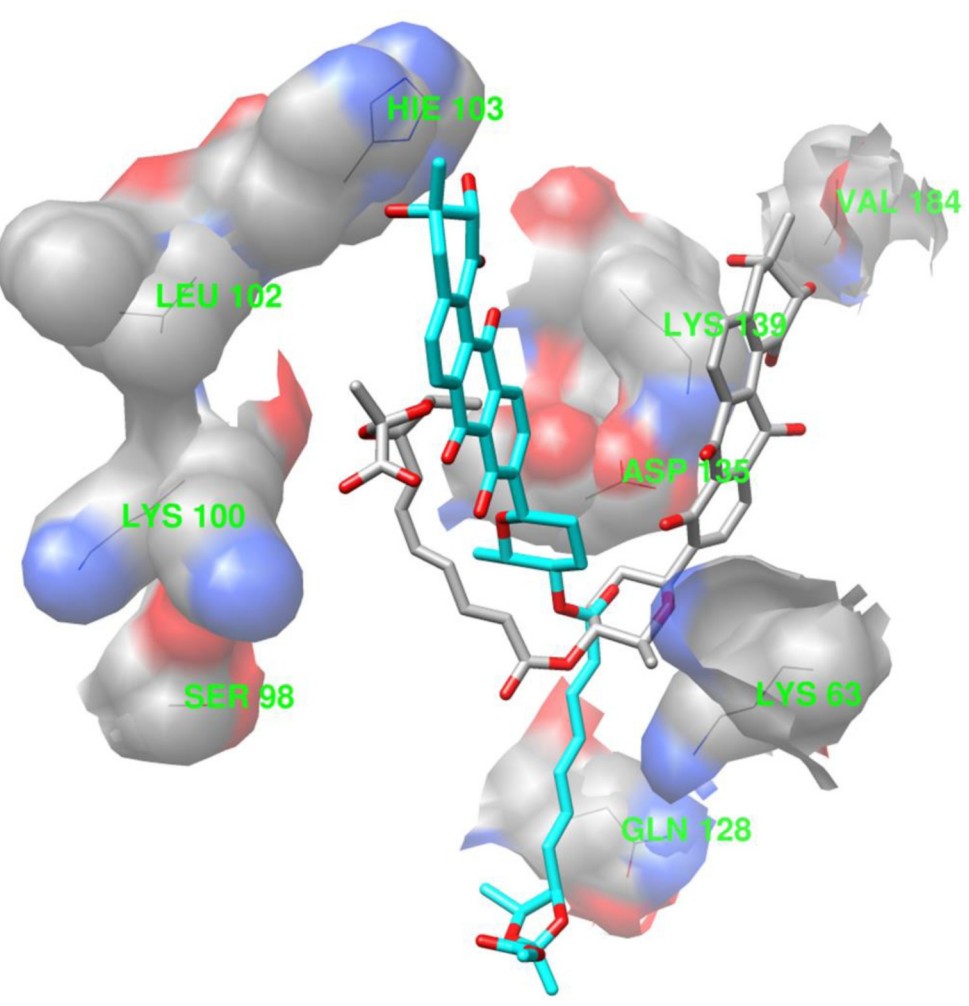

**Fig 8. The final snapshots (35 ns) of ligand–protein complexes in the Molecular Dynamics (MD) simulations.** The ligands are represented as sticks (gray: waldiomycin; cyan: waldiomycin methyl ester). The proteins are represented as wires of constant colors.

binding affinity ($\Delta G_{expt}$) was estimated from binding affinity data using the equation $\Delta G_{expt}$ $\approx -RT\ln IC_{50}$ [43] (Table 2). Interestingly, the binding free energy $\Delta G_{MM/GBSA}$ (waldiomycin, –43.73; waldiomycin ester, –27.75 kcal mol⁻¹) was in accordance with the experimental binding affinity, $\Delta G_{expt}$ (waldiomycin, –26.47 kcal mol⁻¹; waldiomycin ester, –21.85 kcal mol⁻¹) estimated from the $IC_{50}$ values (waldiomycin, 10.2 **μ**M; waldiomycin methyl ester, 75.8 **μ**M). Analyzing the components in the MM/GBSA binding free energies revealed that electrostatic energy was the major contributor to the two binders. For example, $\Delta G_{electrostatic}$ was –17.631 kcal mol⁻¹ for waldiomycin containing carboxyl group, whereas $\Delta G_{electrostatic}$ was –10.093 kcal mol⁻¹ for the waldiomycin containing methyl ester. The strong hydrogen bond formation

**Table 1. Results of pairwise energy decomposition analysis of the high-occupancy hydrogen bonding of residues to the ligands (kcal mol⁻¹).**

| Hk/WalK residues | Lys63 | Ser98 | Lys100 | Leu102 | Hie103 | Gln128 | Asp135 | Lys139 | Val184 |
|---|---|---|---|---|---|---|---|---|---|
| **Waldiomycin** | –1.093 | –0.04 | –4.371 | –0.043 | –0.836 | –0.111 | –2.722 | –3.29 | –4.136 |
| **Waldiomycin ester** | –1.575 | –0.088 | –0.096 | –0.025 | –0.092 | –1.064 | –1.497 | –2.536 | –1.345 |

**Table 2. Calculated binding free energies in comparison with experimental data (kcal mol$^{-1}$ [a]).**

| Ligand | $\Delta G_{vdw}$ | $\Delta G_{elec}$ | $\Delta G_{polar}$ [b] | $\Delta G_{Surf}$ [c] | $\Delta G_{MM/GBSA}$ | $\Delta G_{Expt}$ [d] |
|---|---|---|---|---|---|---|
| Wald | −58.144 | −17.631 | 40.174 | −8.132 | −43.733 | −26.468 |
| Wald ester | −47.868 | −10.093 | 37.403 | −7.195 | −27.753 | −21.849 |

[a] Average of 1000 frames

[b] Whole electrostatic contribution: $\Delta G_{elec} = \Delta G_{electrostatic} + \Delta G_{polar}$

[c] Whole nonpolar contribution: $\Delta G_{np} = \Delta G_{vdw} + \Delta G_{surf}$

[d] $\Delta G_{expt} = RT\ln IC_{50}$, T = 277 K, R = 8.314 JK$^{-1}$mol$^{-1}$

between waldiomycin and the key residue Lys100 was recognized and postulated to be the reason for the 7.4-fold higher potency of waldiomycin than that of its ester analog.

## Conclusions

In a recent study, an inhibitors of *S. aureus* HK/WalK enzyme, waldiomycin, was reported to have 7.4-fold higher potency than its other inhibitor, waldiomycin ester. The two inhibitors have the same structures but differ with regard to only one pharmacophore. Knowledge of the compounds' binding free energies is crucial to evaluate its potential antibacterial activity. The tertiary structures of *S. aureus* HK/WalK are crucial to understanding HK/WalK–ligand interactions for evaluating potential anti-*S. aureus* activity. The amino acid residue Lys100 was crucial for hydrogen bonding with waldiomycin at the binding site and possibly played an essential role in the enzyme activity. The results of our work on binding interactions between the ligands and *S. aureus* HK/WalK would be useful for further studies, including the designing of more efficient antibacterial agents that target *S. aureus* HK/WalK.

## Supporting information

**S1 Fig. Ramachandran plot of the optimized *S. aureus* HK/WalK structure.**
(PDF)

**S2 Fig. Ramachandran plot of the initial homology modeled *S. aureus* HK/WalK structure.**
(PDF)

**S3 Fig.** The co-crystallized ACP (from 5c93.pdb, colored magenta) and the redocked ACP structure (colored forest green), superimposed inside the binding site of 5c93.
(TIF)

**S4 Fig. The 10 top-scored conformations of the waldiomycin in mol2 format.**
(TIF)

**S5 Fig. The 10 top-scoring conformations of the waldiomycin methyl ester in mol2 format.**
(TIF)

**S1 File. The docking score and hydrogen bond information of the top 10 scored conformations of waldiomycin.**
(INFO)

**S2 File. The docking score and hydrogen bond information of the top 10 scored conformations of waldiomycin methyl ester.**
(INFO)

## Acknowledgments

The authors extend their appreciation to the Deanship of Scientific Research at King Saud University.

## Author Contributions

**Conceptualization:** Awwad Radwan.

**Data curation:** Awwad Radwan.

**Formal analysis:** Awwad Radwan.

**Methodology:** Awwad Radwan.

**Project administration:** Gamal M. Mahrous.

**Resources:** Gamal M. Mahrous.

**Software:** Awwad Radwan.

**Visualization:** Gamal M. Mahrous.

**Writing – original draft:** Awwad Radwan.

**Writing – review & editing:** Awwad Radwan.

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
