## [Decision Letter · Decision Letter 0]

31 Mar 2020

PONE-D-20-05993

Docking studies and molecular dynamics simulations of the binding characteristics of waldiomycin and its methyl ester analogue to histidine kinase of Staphylococcus aureus

PLOS ONE

Dear Dr Awwad Radwan,

Thank you for submitting your manuscript to PLOS ONE. After careful consideration, we feel that it has merit but does not fully meet PLOS ONE’s publication criteria as it currently stands. Therefore, we invite you to submit a revised version of the manuscript that addresses the points raised during the review process.

I fully agree with the comments indicated by the referees. All the points indicated need to be addressed in detail. Additionally, the authors must consider and discuss accordingly the possibility that the ester may be a proform of the carboxylic acid. Therefore, it is important to address the importance of having a negatively charged ligand for recognition, as noted in the comments. This is quite often. 

We would appreciate receiving your revised manuscript by April 30th. To enhance the reproducibility of your results, we recommend that if applicable you deposit your laboratory protocols in protocols.io, where a protocol can be assigned its own identifier (DOI) such that it can be cited independently in the future. For instructions see: http://journals.plos.org/plosone/s/submission-guidelines#loc-laboratory-protocols

We look forward to receiving your revised manuscript.

Kind regards,

Concepción Gonzalez-Bello, Ph.D.

Academic Editor

PLOS ONE

Journal Requirements:

Additional Editor Comments (if provided):

Reviewers' comments:

Reviewer's Responses to Questions

**Comments to the Author**

1. Is the manuscript technically sound, and do the data support the conclusions?

Reviewer #1: Yes

Reviewer #2: Partly

2. Has the statistical analysis been performed appropriately and rigorously? 

Reviewer #1: Yes

Reviewer #2: Yes

3. Have the authors made all data underlying the findings in their manuscript fully available?

Reviewer #1: Yes

Reviewer #2: Yes

4. Is the manuscript presented in an intelligible fashion and written in standard English?

Reviewer #1: Yes

Reviewer #2: Yes

5. Review Comments to the Author

Reviewer #1: The authors of this paper study the binding properties of waldiomycin and its methyl ester analog to histidine kinase (HK) of Staphylococcus aureus. To this purpose, the 3D structure of the catalytic domain of S. aureus HK was first generated using a homology model. Then, they study the docking of the two molecules and perform MD simulations on the more stable predicted complexes. Finally, the binding free energies of the two derivatives were calculated using the generalized Born surface area approach (MM-GBSA). According to the calculations, the higher affinity displayed by waldiomycin relative to its methylated analog can be explained because the carboxylate group of the former is engaged in a hydrogen bond with Lys100 of the protein. This stabilizing interaction is not possible in the methyl ester variant.

The work is well conducted, and the reviewer believes that it can deserve publication in PLOS ONE.

However, before it is published, authors should clarify the followings:

1) The homology model is based on a protein that presents only an identity of 36%. This result could lead to erroneous conclusions. This point should be explicitly discussed in the manuscript.

2) Is there any experimental evidence that supports the significance of Lys100 for the recognition of waldiomycin?

3) The MM-GBSA approach does not consider the entropy associated with the binding. This point should be discussed in the paper.

Minor points:

1) Page 6. Lines 142-144. The following paragraph should be rewritten: 'Using the TIP3P water solvent model, the protein molecule was included in a water box with 10 Å thickness and using sodium ions were as counter ions and using explicit solvation'.

2) Page 7. Lines 163-164. The sentence is not correct. The GAFF force field is applied only for the ligands and not for the complexes. In this regard, what is the approach used to calculate the partial charges of the ligands (BCC, HF/6-31G*, etc.)? What is the starting structure used for these ligands? The X-ray structure of waldiomycin has been previously reported (deposition number: CCDC 892997)

3) The structure of the best ten candidates derived from the docking calculations, together with their corresponding score, should be included in the Supporting Information.

4) The MD simulations were done for a short time (35 ns). What is the reason for using this simulation time?

5) Page 12. The interaction between the ligands and the receptor should be explained in detail (hydrogen bonding, CH/p, etc.).

6) Figure 2: the font used is tiny and difficult to read. Why are the authors showing two different binding sites?

7) Figure 3: the double bond in one of the aromatic rings of both molecules is not adequately represented.

8) Figure 5: the number are difficult to read. In Figure 5a, is it true that the red line corresponds to all atoms of the protein? There is a typo in the Figure caption: 'toms' should be 'atoms’.

9) Figure 6: The labels are difficult to read.

10) Figure 7: An interaction diagram should be added for both complexes.

11) Combine Figure 8 and 9 and use the same color code as the one used in Figure 9 to show the RMSD of the methylated derivative.

Reviewer #2: Bacterial histidine kinases (HK) are interesting targets for the design of antibacterial agents. The authors here report the computational studies of the binding to the S. aureus HK/Walk domain of two known inhibitors: waldiomycin and its methyl ester analogue. The interest of the research is based in the difference in the potency of these two ligands: waldiomcyin presents 7.4‐ fold stronger potency for over waldiomycin ester. The authors provide a 3D perspective about their biding modes correlated with this affinity difference. They propose a homology model of the S. aureus HK enzyme and use it for docking and MD simulations. The computational work has been conducted in a professional manner. The here reported studies are relevant in the context of finding novel antimicrobial agents and providing new insights into the SAR of waldiomycin as S. aureus HK inhibitor.

However, the results are not sufficiently well detailed and discussed and some additional studies are missed.

The antimicrobial activity (MIC) of waldiomycin is reported in ref. 20 (Igarashi et al., 2013). At the conclusions, the authors state that “waldiomcyin presents 7.4‐ fold stronger potency for over waldiomycin ester”. This statement should be justified in the main text and also the context where “potency” is used. In any case, the concepts of “affinity” and “potency” are mixed along the manuscript. This should be revised and clarified.

Line 87. “Unfortunately, few inhibitors of TCS have been described against S. aureus, a major pathogen with high virulence factor”. A reference should be cited here. Which inhibitors are these? They should be docked in order to compare with waldiomycin. For example, the compounds cited in ref. 20 (Igarashi et al., 2013) could be good examples to be docked and compared with waldiomycin.

Also the lack or low binding of waldiomycin towards human kinases could be studied (docking) as control, in order to avoid secondary/toxic effects. Lys100 was found to be crucial residue for S aureus HK binding of waldiomcin. Is this Lys present in human kinases?

The discussion about the docking poses and the results about the binding pose after the MD simulation should be enriched: more details about the interactions, pockets and residues that are important for the binding.

It is not clearly mentioned which is the protonation state considered for waldiomycin (protonation state under physiological condition). Is it docked as carboxylate or as carboxylic acid? The authors mention the H-bond with Lys100 (from the figures, it seems that with the side chain ammonium). This interaction drives the carboxylic/carboxylate group towards the Lys100 ammonium, while in the case of the waldiomycin methyl ester is not clear which interactions are not stable, and which new interactions are stablished in the new binding pose after the simulation (why now the new interactions do stabilize the new pose).

Why does waldiomycin methyl ester migrates to other binding pocket? Was this pocket reached by any predicted docked pose? This compound has a reported low affinity but it remains bound during the simulation. What does the energetics analysis say about the interactions? The contributions to the binding free energy from each residue should be analyzed from the MM-GBSA and discussed in the light of the new binding pose for the waldiomycin methyl ester.

Can the authors deepen into the required interactions for a ligand to be a good S. aureus HK inhibitors?

Minor corrections

-Line 279-284. Refs 37 and 38 referring to limitations in ligand-protein docking (2012 and 2002) and ref 39 as publication to illustrate advances in MD simulations (2011) are obsolete. More recent references should be cited.

-FIG5.

(a) Scale of the time should be adapted at the actual time of the simulation (0 to 30 ns?).

(b) The residue numbers should (x-axis) should be cut before 250. A label should be added indicating this corresponds to residue numbering.

-FIG.6 should go to SI. It´s a calculation made as validation, it´s not a main result of the work.

-Typos:

Line 90- “the 3D structure is unavailable in protein databank”.

It probably should say “the 3D structure is unavailable in the protein databank”. The PDB web page should be added as reference.

Additional work is required, which I think is necessary for the sake of sound conclusions, impact of the work and usefulness to the scientific community. In view of the comments noted above, I would recommend its acceptance for publication in PLOS ONE only when the corrections have been addressed.

6. PLOS authors have the option to publish the peer review history of their article (what does this mean?). If published, this will include your full peer review and any attached files.

Reviewer #1: No

Reviewer #2: No

---

## [Author Response · Author response to Decision Letter 0]

24 Apr 2020

Dear sir 

I'm so thankful for your valuable comments and suggestions.

Addition work was performed and included in the manuscript based on the reviewer comment. The work includes hydrogen bond analysis cpptraj analysis and the pairwise energy decomposition analysis using mmpbsa calculations in Amber18.

The revised submission includes:

The response for reviewer file includes point by point answer of the reviewer comments.

Mansuscript with track changes.

There is also a clear manuscript file after language edition through editage service.

Certificate of editing from editage service is included.

Supporting information are uploaded as zip compressed archive file.

Wit my grateful appreciation to your help and communications.

---

## [Decision Letter · Decision Letter 1]

18 May 2020

PONE-D-20-05993R1

Docking studies and molecular dynamics simulations of the binding characteristics of waldiomycin and its methyl ester analog to Staphylococcus aureus histidine kinase

PLOS ONE

Dear Dr. Radwan,

Thank you for submitting your manuscript to PLOS ONE. After careful consideration, we feel that it has merit but does not fully meet PLOS ONE’s publication criteria as it currently stands. Therefore, we invite you to submit a revised version of the manuscript that addresses the points raised during the review process.

The manuscript requires to address the points highlighted by the two referees to be suitable for publication. 

We would appreciate receiving your revised manuscript by May 25th 2020. To enhance the reproducibility of your results, we recommend that if applicable you deposit your laboratory protocols in protocols.io, where a protocol can be assigned its own identifier (DOI) such that it can be cited independently in the future. For instructions see: http://journals.plos.org/plosone/s/submission-guidelines#loc-laboratory-protocols

We look forward to receiving your revised manuscript.

Kind regards,

Concepción Gonzalez-Bello, Ph.D.

Academic Editor

PLOS ONE

Reviewers' comments:

Reviewer's Responses to Questions

**Comments to the Author**

1. If the authors have adequately addressed your comments raised in a previous round of review and you feel that this manuscript is now acceptable for publication, you may indicate that here to bypass the “Comments to the Author” section, enter your conflict of interest statement in the “Confidential to Editor” section, and submit your "Accept" recommendation.

Reviewer #1: All comments have been addressed

Reviewer #2: All comments have been addressed

2. Is the manuscript technically sound, and do the data support the conclusions?

Reviewer #1: Yes

Reviewer #2: Yes

3. Has the statistical analysis been performed appropriately and rigorously? 

Reviewer #1: Yes

Reviewer #2: Yes

4. Have the authors made all data underlying the findings in their manuscript fully available?

Reviewer #1: Yes

Reviewer #2: Yes

5. Is the manuscript presented in an intelligible fashion and written in standard English?

Reviewer #1: Yes

Reviewer #2: Yes

6. Review Comments to the Author

Reviewer #1: All my comments have been addressed, and the manuscript is now acceptable for publication.

Minor points:

1) Units of the grid score should be indicated in both the main text and Supporting Information.

2) Units of pairwise energy decomposition should be shown in Table 1 and the main text. Is this energy given in Kcal or Kcal/mol? (see main text: 'range value -2.722 to -4.371 Kcal (Table 1)'

Reviewer #2: The authors have successfully addressed all the concerns and corrections.

Minor correction:

Line 301: Units should be specified. And probably E values can be rounded to two decimals (this is sufficiently precise in this order of magnitude).

7. PLOS authors have the option to publish the peer review history of their article (what does this mean?). If published, this will include your full peer review and any attached files.

Reviewer #1: No

Reviewer #2: No

---

## [Author Response · Author response to Decision Letter 1]

18 May 2020

Dear Sir

 The corrections were done in view of the reviewer comments. 

The point by point answer of the reviewer comments are listed below:

6. Review Comments to the Author

Reviewer #1: All my comments have been addressed, and the manuscript is now acceptable for publication.

Minor points:

**1) Units of the grid score should be indicated in both the main text and Supporting Information.

Answer:

>>Page 14, line 284,285: The unit, “kcal mol–1 “, was included in the main test; and also in S6 and S7 files of the Supporting Information.

**2) Units of pairwise energy decomposition should be shown in Table 1 and the main text. Is this energy given in Kcal or Kcal/mol? (see main text: 'range value -2.722 to -4.371 Kcal (Table 1)'

Answer:

 >>Page 17, line 348. The unit was changed into “kcal mol-1). 

Reviewer #2: The authors have successfully addressed all the concerns and corrections.

Minor correction:

Line 301: Units should be specified. And probably E values can be rounded to two decimals (this is sufficiently precise in this order of magnitude).

 Answer:

>>Page 18, lines 362, 363. E values were rounded to two decimals and units “kcal mol-1” were specified.

---

## [Editor Report · Decision Letter 2]

21 May 2020

Docking studies and molecular dynamics simulations of the binding characteristics of waldiomycin and its methyl ester analog to Staphylococcus aureus histidine kinase

PONE-D-20-05993R2

Dear Dr. A. Radwan,

We are pleased to inform you that your manuscript has been judged scientifically suitable for publication and will be formally accepted for publication once it complies with all outstanding technical requirements.

With kind regards,

Concepción Gonzalez-Bello, Ph.D.

Academic Editor

PLOS ONE
---

## [Editor Report · Acceptance letter]

26 May 2020

PONE-D-20-05993R2 

Docking studies and molecular dynamics simulations of the binding characteristics of waldiomycin and its methyl ester analog to Staphylococcus aureus histidine kinase 

Dear Dr. Radwan:

I am pleased to inform you that your manuscript has been deemed suitable for publication in PLOS ONE. Congratulations! Your manuscript is now with our production department. 

With kind regards,

on behalf of

Dr. Concepción Gonzalez-Bello 

Academic Editor

PLOS ONE